# Response of the Nordic Seas to the 2-6 February 2020 Marine Cold Air Outbreak in the GLORYS12 Ocean Reanalysis

Svenya Chripko<sup>1</sup>, Thomas Spengler<sup>1</sup>, Stefanie Semper<sup>1</sup>, and Kjetil Våge<sup>1</sup>

<sup>1</sup>Geophysical Institute, University of Bergen, and Bjerknes Centre for Climate Research, Bergen, Norway.

**Correspondence:** Svenya Chripko (svenya.chripko@uib.no)

#### Abstract.

Marine Cold Air Outbreaks (MCAOs) play a crucial role in wintertime water mass transformation in the Nordic Seas. However, due to the spatio-temporal variability of atmospheric forcing and lateral ocean transport, the processes by which MCAOs influence the ocean remain unclear. Using the eddy-resolving ocean and sea ice reanalysis GLORYS12, we investigate the mechanisms driving the ocean response over the Nordic Seas to the particularly intense 2–6 February 2020 MCAO event. To assess the impact of the MCAO on the ocean, we quantify the contributions of the mean surface turbulent heat flux relative to the mean change in ocean heat content during the event. The western part of the Nordic Seas (Greenland Sea and northern interior Iceland Sea) was primarily affected by the air-sea heat exchanges, with an overall mixed layer cooling by approximately  $0.02 \, ^{\circ}\text{C} \cdot \text{day}^{-1}$  during the event in the interior Greenland Sea and a deepening of more than 30 m·day<sup>-1</sup> in some areas. In the eastern part (Norwegian Sea), on the other hand, the air-sea heat exchanges were masked by stronger lateral oceanic heat transport, with a cooling or warming of an order of magnitude higher. In the interior part of northern Iceland Sea, the mixed-layer depth increased by approximately 5 m·day<sup>-1</sup>, while it decreased near the boundary current in the western Iceland Sea by approximately 8 m·day<sup>-1</sup> concomitantly with a shoaling of the warm Atlantic-origin water mass.

## 1 Introduction

The Nordic Seas (Greenland, Iceland, and Norwegian Seas; Fig. 1) play an important role in global heat transport and the Atlantic meridional overturning circulation (AMOC; Gebbie and Huybers, 2010), with warm saline Atlantic surface waters entering the eastern part of the Nordic Seas and progressively transformed into cold, dense Atlantic-origin Water due to strong air-sea heat loss (Eldevik et al., 2009; Våge et al., 2011; Latarius and Quadfasel, 2016). This water mass then flows southward within the East Greenland Current (EGC; Håvik et al., 2017), feeding the lower limb of the AMOC when exported as dense overflow water through the Denmark Strait (Østerhus et al., 2019). Another contribution to this overflow is Arctic-origin Water, which is produced through buoyancy fluxes yielding wintertime open-ocean convection in the interior Greenland and Iceland Seas (Moore et al., 2015; Mastropole et al., 2017; Våge et al., 2015). Water mass transformation in the Nordic Seas is thus highly dependent on atmospheric forcing, with 60%-80% of the air-sea heat exchange occurring during few but intense marine cold air outbreaks (MCAOs; Papritz and Spengler, 2017). While the heat loss during MCAOs has a crucial impact on the ocean mixed layer in the Nordic Seas (Renfrew et al., 2023; Huang et al., 2021; Svingen et al., 2023), the local impact of the MCAO

50

surface forcing can be offset by lateral advection in the ocean. To further explore these complexities, we use the high-resolution Mercator Ocean global oceanic and sea ice GLORYS12 reanalysis (Lellouche et al., 2021) and quantify the oceanic response in the Nordic Seas to the intense MCAO event 2–6 February 2020 during the COMBLE campaign (Geerts et al., 2022).

MCAOs are wintertime weather events, where polar air masses are advected from sea ice or cold land surfaces towards the relatively warmer open ocean (Papritz and Spengler, 2017). They are the manifestation of a prolonged period of atmospheric thermal instability, in which large amounts of heat are extracted from the ocean and transferred to the atmospheric boundary layer. This leads to atmospheric mixing and the typical convective cloud streets (see Geerts et al., 2022, and references therein). MCAOs can be modulated by orographically-driven mesoscale wind systems such as barrier or katabatic winds (Gutjahr et al., 2022), or by polar lows formed at the sea ice edge (Gutjahr and Mehlmann, 2024; Geerts et al., 2022). MCAO events typically last 2-4 days in the Iceland Sea and Fram Strait region, with some of them occasionally reaching lifetimes of more than a week (Harden et al., 2015; Terpstra et al., 2021). The largest air-sea turbulent heat fluxes generally occur rapidly after the onset of MCAOs, with mean values reaching approximately 300 W·m<sup>-2</sup> within the first 6 hours (Papritz and Spengler, 2017).

Ocean observations have been used to analyze the effects of seasonal or interannual variations of surface heat fluxes on the ocean mixed-layer in the Irminger Sea (Våge et al., 2008), Greenland and Iceland Seas (Moore et al., 2015; Våge et al., 2015, 2018), and Norwegian Sea (Huang et al., 2023; Sætra et al., 2008; Isachsen et al., 2013). More recent studies have also investigated the impact of specific MCAO events and found that the response of the mixed-layer depth (MLD) to MCAOs is strongly dependent on the timing of the events, with early winter events mainly cooling the mixed layer, while late winter events mainly deepening the mixed layer (Svingen et al., 2023; Semper et al., 2025). Huang et al. (2021) and Renfrew et al. (2023) focused on a single MCAO early 2018 in the Iceland Sea (Renfrew et al., 2019), highlighting a strong spatial variability in the oceanic response, with mixed layer cooling in the interior of the Iceland Sea and mixed layer warming in the boundary current region. The former was mainly attributable to atmospheric forcing, whereas the latter was hypothesized to be due to lateral advection of the warm Atlantic-origin Water (Huang et al., 2021; Renfrew et al., 2023). We further disentangle the highlighted spatial variability in the oceanic response to one specific intense MCAO event in the Nordic Seas using the GLORYS12 reanalysis.

The GLORYS12 reanalysis performs well in representing the physical ocean state at an eddy-resolving resolution (Lellouche et al., 2021), with approximately 5 km horizontal resolution in the Nordic Seas. The reanalysis is forced by the European Centre for Medium-Range Weather Forecast (ECMWF) global atmospheric reanalysis ERA5 at the surface (Hersbach et al., 2020), which resolves relevant meteorological phenomena, such as MCAOs and polar lows, at an unprecedented level (Meyer et al., 2021). The GLORYS12 reanalysis thus provides an opportunity to develop an understanding of the mesoscale impacts of MCAOs on the ocean mixed layer in the different regions of the Nordic Seas. Our focus is on the extreme winter season 2020 (Dahlke et al., 2022), using the 2–6 February 2020 MCAO event. This specific MCAO event represents one of the two strongest February events in terms of daily peak intensity for the period 1979–2020 in the Fram Strait region (Dahlke et al., 2022). It was also detected at the Bear Island and Andenes observational sites deployed during the COMBLE campaign (Geerts et al., 2022) and affected the entire Nordic Seas.

**Figure 1.** Schematic of the Nordic Seas (Greenland, Iceland, and Norwegian Seas) illustrating the bathymetry (blue shading) and the main currents (arrows). The red arrows indicate warm and salty surface Atlantic waters entering the Nordic Seas, while the green arrows indicate colder and denser waters. The currents in the Norwegian Sea and near the Norwegian coast are, respectively, the Norwegian Atlantic Current (NAC, slope and frontal branches) and the Norwegian Coastal Current (NCC). The East Greenland Current (EGC) is situated in the western part of the Greenland and Iceland Seas and leaves the Nordic Seas through the Denmark Strait. The North Icelandic Irminger Current (NIIC) flows eastward along the north Icelandic coast. The two thick black lines show the defined Greenland 75°N (16°W-0°E) and Iceland Sea 71°N (21°W-10°W) sections.

## 60 2 Data and methods

# 2.1 Atmosphere and ocean reanalyses

We investigate the MCAO from 31 January 2020 until 6 February 2020. For the atmosphere, we use the European Center for Medium-Range Weather Forecasts (ECMWF) global reanalysis ERA5 (Hersbach et al., 2020), with a 0.25° grid resolution. We use hourly output for sea level pressure, surface temperature, 10-meter wind, as well as 850-hPa temperature and wind, and calculate daily mean surface fluxes from 3-hourly outputs for surface sensible and latent heat flux as well as net surface shortwave and longwave radiations. ERA5 has been shown to perform well over the ice-free ocean in the Iceland and Greenland

Seas, though surface temperature, wind speed, and sensible heat flux are less accurate over the marginal ice zone (Renfrew et al., 2021).

To analyze the ocean response to the MCAO, we use the Mercator Ocean global eddy-resolving reanalysis GLORYS12 (Lellouche et al., 2021), which has a  $1/12^{\circ}$  horizontal resolution (around 5 km in the Nordic Seas) and 50 vertical levels. GLO-RYS12 covers the period 1993 onward and is based on the NEMO ocean and sea ice model (Gurvan et al., 2017). The model is forced at the surface by the ECMWF ERA-Interim atmospheric reanalysis (Dee et al., 2011) and ERA5 from 2019. GLO-RYS12 assimilates sea level, sea surface temperature, sea ice concentration, as well as temperature and salinity vertical profiles (https://documentation.marine.copernicus.eu/PUM/CMEMS-GLO-PUM-001-030.pdf). We use daily mean three-dimensional potential temperature, practical salinity, northward and eastward ocean current velocity, and the two-dimensional mixed-layer depth (MLD) as well as sea ice concentration (SIC). The MLD in GLORYS12 is defined as the depth where the density increase, compared to 10-m depth, corresponds to a temperature decrease of  $\Delta T = 0.2^{\circ}$ C (https://documentation.marine.copernicus.eu/PUM/CMEMS-GLO-PUM-001-030.pdf). We compute the Conservative Temperature, Absolute Salinity, and potential density referenced to the surface following the TEOS-10 standard (IOC et al., 2010). If not specified, "temperature", "salinity", and "density" hereafter refer to "Conservative Temperature", "Absolute Salinity", and "potential density", respectively.

GLORYS12 performs well for the low-frequency variability of sea ice extent (Lellouche et al., 2021), the interannual variability of hydrographic properties in the Irminger Sea (Sterl and de Jong, 2022), and the long-term mean current structure in the North Atlantic (Verezemskaya et al., 2021) and Nordic Seas (Huang et al., 2023). Huang et al. (2023) found a small offset of 0.02 m·s<sup>-1</sup> between the 2005-2018 mean current velocities in GLORYS12 and observations for the Norwegian Atlantic Current and East Greenland Current, comparing depth-means until 1000 m depth. GLORYS12 also shows good agreement with observations regarding water masses and volume transport in the Nordic Seas (Årthun et al., 2025).

#### 2.2 MCAO index

To characterize the spatial and temporal evolution of the MCAO and its intensity, we calculate the instantaneous MCAO index defined as the potential temperature difference between the ocean surface and the air at 850 hPa,  $\theta_{SST} - \theta_{850 \ hPa}$  (Papritz et al., 2015; Kolstad, 2017; Polkova et al., 2019, 2021; Meyer et al., 2021).

#### 2.3 Ocean heat budget and contribution of surface heat fluxes

We calculate the ocean heat content (H) between the surface and the ocean bottom, using:

$$H = c_p \int_{-D_{b,t}}^{0} \Theta \rho dz , \qquad (1)$$

where  $c_p$  is the constant heat capacity of seawater,  $D_{bot}$  the ocean bottom depth,  $\Theta$  the Conservative Temperature, and  $\rho$  the in-situ density. The local tendency,  $\partial H/\partial t$ , integrated over one day d, represents the daily change in the column-averaged ocean heat content ( $\Delta H = \int_0^d \frac{\partial H}{\partial t} dt$ ). Note that we integrate the ocean heat content in Eq. 1 down to the ocean bottom to simplify

100

the heat budget conservation equation as in Roberts et al. (2017):

$$\frac{\partial H}{\partial t} = -SFC + CONV \,, \tag{2}$$

where SFC is the upward total net surface heat flux and CONV is the ocean heat flux convergence due to lateral advection processes. Vertical advection terms cancel out when integrating to the ocean bottom. SFC is the sum of the upward net shortwave and longwave radiation flux  $Q_{RAD}$  together with the upward net surface turbulent heat flux  $Q_{THF}$ . The contribution of  $Q_{RAD}$  is negligible compared to  $Q_{THF}$  (not shown) and the total net surface heat flux is mainly explained by the net surface turbulent heat flux.

To isolate the direct impact of the MCAO surface forcing on the ocean properties during the 5 days of the event, we calculate the ratio (r) between the 5-day mean surface turbulent heat flux penetrating the ocean  $-Q_{THF}$  and the 5-day mean ocean heat content change for the duration of the MCAO, in the Nordic Seas. A ratio  $r \approx 1$  indicates that surface heat fluxes contribute more or less exclusively to the ocean heat content change, i.e., where we consider the MCAO to have a significant impact on the ocean mixed-layer properties. A ratio r > 1 indicates that other processes are counteracting the contribution of surface heat fluxes and if  $0 \le r \le 1$ , surface heat fluxes are not strong enough to explain the ocean heat content change and other processes contribute to changes in H. A negative ratio r < 0, on the other hand, indicates that surface heat fluxes have the opposite sign compared to changes in H, indicating that changes in H are dominated by other processes. As introduced in Eq. 2, we assume that other processes are mainly associated with lateral ocean heat advection due to ocean currents and eddies.

#### 3 Atmospheric and sea ice conditions during the MCAO

On 31 January 2020, prior to the MCAO, the synoptic situation was characterised by strong north-northeasterly surface winds along the sea ice edge east of Greenland and a low in the Norwegian Sea (Fig. 2a). At 850 hPa, the wind field was similar in the Iceland Sea, but significantly weaker in Fram Strait (Fig. 2c). The polar low existed until 1 February (not shown) and was detected in northern Norway during the COMBLE campaign (Geerts et al., 2022). The strong winds along the east Greenland sea ice edge were accompanied by large upward surface heat fluxes (Fig. 2a), dominated by sensible heat fluxes along the sea ice edge (Fig. 2b). The cold northerlies behind the low were probably associated with favorable conditions for the development of the MCAO (Fig. 2c).

On 3 February, the wind shifted to north-westerlies, especially in the northern part of the Nordic Seas, both at the surface (Fig. 2d) and 850 hPa (Fig. 2f). These conditions were responsible for extremely strong heat fluxes from the ocean to the atmosphere, reaching around 600 W·m<sup>-2</sup> on 31 January (Fig. 2a) and locally more than 1000 W·m<sup>-2</sup> on 3 February (Fig. 2d) in the Fram Straight, with the largest contribution from sensible heat fluxes (Fig. 2e). According to Papritz and Spengler (2017), this MCAO event classifies as a "very strong" intensity event, with the MCAO index reaching more than 20 °C in the Fram Strait region, approximately 15 °C in the Greenland Sea, and below 10 °C in the Iceland and Norwegian Seas, and with temperatures at 850 hPa varying from -34 °C to -6 °C in these areas (Fig. 2f). We hence consider this day as the peak of the 2-6 February MCAO event, with the highest surface fluxes and MCAO index (not shown). The flow configuration was rather stable

**Figure 2.** Synoptic situation during the MCAO in ERA5. (a), (d), and (g): total surface heat flux in shading (latent, sensible, longwave and shortwave radiation, W⋅m<sup>-2</sup>), sea level pressure in blue contours (970 to 1015 hPa every 5 hPa), 10 meter-wind vector in black. (b), (e), and (h): sensible heat flux in shading and latent heat flux in thick purple contours at 0, 100, and 200 W⋅m<sup>-2</sup>. The 500-, 1000-, 2000-, and 3000-m isobaths are shown as thin black contours. (c), (f), and (i): temperature at 850 hPa in shading (°C; the thin grey dotted line indicates the 0°C contour), wind vector at 850 hPa, and MCAO index in orange contours (°C). The first row corresponds to 31/01/2020 (before MCAO), the second row to 03/02/2020 (peak MCAO), and the last row to 06/02/2020 (last day MCAO). Every quantity is instantaneous at 12:00 UTC, except the heat fluxes which are averaged over the considered day. The magenta contour line corresponds to the sea ice edge at 15 % sea ice concentration in GLORYS12.

from the onset on 2 February until 5 February, with a larger extraction of sensible heat compared to latent heat, especially in the region off the sea ice edge, where the air-sea temperature contrast was largest (not shown).

On 6 February, the last day of the MCAO, the flow was perturbed by a cyclone over the Iceland Sea (Fig. 2g), with the wind changing direction, advecting a warmer air mass from the south across the south-western part of the Nordic Seas (Fig. 2i). In the north-eastern part, remnants of the northwesterlies were still associated with a moderate MCAO index (< 10 °C, Fig. 2i) and strong surface fluxes (300–450 W·m<sup>-2</sup>, Fig. 2g).

This particularly intense MCAO not only affected the ocean-atmosphere heat exchanges over the Nordic Seas, but also impacted the sea ice distribution. During the event, the sea ice edge around Greenland expanded noticeably offshore (Figs.2a,d,g), with a maximum extent on 4–5 February, followed by a slight retreat (not shown). The sea ice volume also increased along the sea ice edge, but decreased near the coastline of Greenland (Fig. S1). A structure reminiscent of the Odden ice tongue (Comiso et al., 2001) formed around 73°N–11°W (Fig. 2d,g), and was present from 2 to 9 February (not shown). Hence, sea ice formation, melting, and advection might have occurred concurrently and affected the ocean mixed-layer properties locally during the MCAO event. However, we here focus on the impact of the air-sea heat exchange on the ocean during the event.

## 4 Ocean response to the MCAO

## 4.1 Ocean mixed-layer changes over the Nordic Seas

Prior to the event, the Nordic Seas were characterized by west-east gradients in surface temperature, salinity, and density, with colder, less saline, and denser water masses west of the Jan Mayen, Mohn, and Knipovich Ridges (31 January 2020, Fig. 3a,b,c and Fig. 1). An exception is found along the Norwegian shelf, where the mixed layer was less saline, due to fresh water (< 34.5 g·kg<sup>-1</sup>) transported northward by the Norwegian Coastal Current (Furevik et al., 2002, Fig. 1). The Icelandic coast is another exception, where warmer and more saline surface waters are carried poleward by the North Icelandic Irminger Current (Casanova-Masjoan et al., 2020; Semper et al., 2022). The deepest mixed layers were found in the Greenland Sea, exceeding depths of 800 m two days before the MCAO (Fig. 3d). This is consistent with the range of observed MLDs since 2000, when convection has been found to take place at intermediate depth in this region (Strehl et al., 2024). The main currents in the Nordic Seas (Fig. 1) are clearly visible during the MCAO in the GLORYS12 reanalysis (Fig. 3e): the Norwegian Atlantic Front Current, Norwegian Atlantic Slope Current, Norwegian Coastal Current, West Spitsbergen Current, and EGC. Their locations and magnitudes are consistent with the composite of the long-term mean sea surface velocity from satellite observations and from GLORYS12 (Huang et al., 2023).

The ocean surface properties in the Nordic Seas changed markedly during the MCAO between 31 January 2020 and 6 February 2020 (Fig. 4), with the largest changes generally occurring during the peak of the event (not shown). Consistent with the strong surface heat loss (Fig. 2d,g), the surface layer generally cooled in the Nordic Seas during the event (Fig. 4a). There was significant spatial variability, with, for example, a cooling of approximately  $0.1^{\circ}$ C and  $0.4^{\circ}$ C in the western and eastern Greenland Sea, respectively. The cooling locally even exceeded  $1.0^{\circ}$ C in regions with strong ocean currents, such as the West Spitsbergen Current, the EGC north of about  $74^{\circ}$ N, the Jan Mayen Current, and some parts of the Norwegian Atlantic and

Figure 3. Surface ocean conditions before the MCAO (31/01/2020) and mean ocean dynamics during the MCAO. (a) Surface Conservative Temperature ( $^{\circ}$ C), (b) surface Absolute Salinity (g·kg $^{-1}$ ), (c) surface potential density (+ 1000 kg·m $^{-3}$ ), and (d) mixed-layer depth (m) before the MCAO (31/01/2020). (e) Mean surface sea water velocity (m·s $^{-1}$ ) during the MCAO (02/02/2020-06/02/2020). The 500-, 1000-, 2000-, and 3000-m isobaths are shown as thin black contours. The magenta line in panels (a), (b), (c), and (d) corresponds to the sea ice edge at 15 % sea ice concentration. The magenta and cyan lines in panel (e) correspond to the sea ice edges during the first (02/02/2020) and last (06/02/2020) MCAO days, respectively. The two sections in black in panel (e) show the defined Greenland Sea 75 $^{\circ}$ N (16 $^{\circ}$ W-0 $^{\circ}$ E) and Iceland Sea 71 $^{\circ}$ N (21 $^{\circ}$ W-10 $^{\circ}$ W) sections.

**Figure 4.** Difference in surface ocean conditions between the post-MCAO (06/02/2020) and pre-MCAO (31/01/2020) conditions. (a) Surface Conservative Temperature difference ( $^{\circ}$ C), (b) surface Absolute Salinity difference ( $10^{-2} \text{ g} \cdot \text{kg}^{-1}$ ), (c) surface potential density difference ( $10^{-1} \text{ kg} \cdot \text{m}^{-3}$ ), and (d) mixed-layer depth difference (m). The 500-, 1000-, 2000-, and 3000-m isobaths are shown as thin green contours. The magenta and dashed cyan lines, respectively, correspond to the end-MCAO and pre-MCAO sea ice edges at 15 % sea ice concentration. Note the non-linear scale in panels (a), (b), and (c).

Coastal Currents. Several areas, on the other hand, featured a significant warming, exceeding 1.0°C along some of the currents and westward of the Odden-like ice tongue forming during the event (Fig. 4a).

Changes in surface salinity were spatially more variable compared to changes in temperature (compare Figs.4a and b). While there was an increase in salinity in the western part of the interior Greenland Sea ( $> 0.01~{\rm g\cdot kg^{-1}}$ ), there was a decrease in

the eastern part (approximately  $0.01~\rm g\cdot kg^{-1}$ ). The interior of the Iceland Sea became more saline (>  $0.025~\rm g\cdot kg^{-1}$ ), while the sign of the changes was very variable in the Iceland Sea near the sea ice edge. Similarly to the temperature changes, the magnitude of changes in salinity were very large for regions with strong currents (Fig. 4b). Below the sea ice surrounding Greenland, salinity strongly increased, consistent with an increase in sea ice volume (Fig. S1). This was also the case in the area of the Odden-like ice tongue (compare magenta and cyan lines in Fig. 4). On the other hand, surface waters generally became less saline closer to the sea ice edge, where sea ice has expanded into the interior Greenland Sea. From  $70^{\circ}$ N to  $74^{\circ}$ N, this is consistent with sea ice volume partly decreasing close to the Greenland coastline (Fig. S1). From approximately  $74^{\circ}$ N to farther north, the decrease in salinity is, however, linked to the south-eastward extension of Polar Surface Water below the newly formed sea ice (see section 4.3.1).

In line with the overall temperature decrease, the surface density generally increased during the MCAO over the open ocean (Fig. 4c). The increase varies from approximately 0.015 kg·m<sup>-3</sup> in the Greenland Sea, to approximately 0.03 kg·m<sup>-3</sup> in the north-eastern Nordic Seas, exceeding 0.04 kg·m<sup>-3</sup> in the interior of the Iceland Sea, and even exceeding 0.05 kg·m<sup>-3</sup> in regions with strong currents and around the formation of the Odden-like ice tongue. The density changes in the interior of the Iceland Sea are more likely dominated by changes in salinity (Fig. S2), congruent with the higher latent heat fluxes compared to the sensible heat fluxes (Fig. 2e), though due to the strong wind the changes could also be related to mixing with higher salinity waters below. Below sea ice, the density increase and decrease are also closely linked to changes in salinity (compare Figs. 4b and c; and see Fig. S2).

Concurrently with the overall densification of the surface waters away from the sea ice edge, the mixed layer generally deepened within the Nordic Seas during the MCAO. The largest changes occurred in the Greenland Sea, with more than 200 m (i.e., more than 30 m·day<sup>-1</sup>, Fig. 4d). In the interior of the Iceland Sea (east of the Kolbeinsey Ridge, Fig. 1), the MLD increased by more than 100 m in some areas, while it decreased up to 60 m in the region near the sea ice edge (west of the Kolbeinsey Ridge). The strong mixed-layer deepening in the Greenland Sea is consistent with the MLD increase observed in this region during MCAOs in late winter for the period 1999–2009 (38 m·day<sup>-1</sup>, Svingen et al., 2023).

#### 4.2 Contribution of surface heat fluxes

In agreement with the ocean changes in surface temperature and density, the response in ocean heat content was generally larger in regions with strong currents or eddies (Fig. 5a). The largest contrast in the ocean heat content change, reaching an order of magnitude, was between the western part of the Nordic Seas (Greenland and Iceland Seas) and both the northern and eastern parts (Fram Strait and Norwegian Sea). The largest changes in the eastern and northern parts were due to lateral heat transport associated with strong currents and associated eddies (Norwegian Sea and West Spitsbergen Current).

The 5-day mean ocean heat content tendency for the MCAO event is approximately 600 W·m<sup>-2</sup> in the Greenland Sea (e.g., around 75°N), which is approximately twice as much as the ocean heat loss in the Iceland Sea (e.g., around 71°N and 69°N). This difference is consistent with the distribution of total surface heat fluxes for these two regions, with significantly higher surface fluxes in the Greenland Sea (Fig. 2d).

**Figure 5.** Contribution of the surface turbulent heat flux to the mean ocean heat content tendency during the MCAO. (a) Mean ocean heat content tendency (W·m<sup>-2</sup>) during the MCAO (02/02/2020-06/02/2020). (b) Ratio between mean downward surface turbulent heat flux and mean ocean heat content tendency during the MCAO. A ratio near 1 or larger (red colors) indicates areas where the MCAO (downward surface turbulent heat flux) contributes the most to the ocean heat content variations during the event (see section 2.3 for a more detailed description of this ratio). The 500-, 1000-, 2000-, and 3000-m isobaths are shown as thin black contours. The cyan and magenta lines correspond to the sea ice edges (15 % sea ice concentration) during the first (02/02/2020) and last (06/02/2020) MCAO days, respectively. Note the non-linear scale in panel (a). The two thick black lines in panel (b) show the defined Greenland 75°N (16°W-0°E) and Iceland Sea 71°N (21°W-10°W) sections.

The loss of ocean heat content in the western part of the Nordic Seas is largely attributable to the surface turbulent heat loss associated with the MCAO (ratio largely between 0.8 and 2.0, Fig. 5b), in contrast to the eastern part and along the EGC and the West Spitsbergen Current, where lateral ocean heat transport dominates the changes in ocean heat content (ratio around 0 or negative, Fig. 5b). Both surface heat exchange and lateral heat transport explain the changes in ocean heat content in the Fram Strait and in the southern Iceland Sea south of 70°N (more variable ratio with negative and positive values, Fig. 5b).

#### 4.3 Vertical structure of the ocean response in the Greenland and Iceland Seas

We focus on the ocean mixed-layer response to the MCAO in the Greenland Sea and in the northern Iceland Sea, where we have shown that the MCAO has a direct effect on the mixed-layer properties away from sea ice and the EGC. In the Greenland Sea, we select a section across the east Greenland shelf, slope, and basin at 75°N, which is commonly used for hydrographic and geochemical observations (Brakstad et al., 2023; Jeansson et al., 2023; Olsen et al., 2024). In the Iceland Sea, we select

a section at 71°N, as in Håvik et al. (2017) and Våge et al. (2018). Farther south, the effect of the MCAO surface forcing on the ocean is masked by lateral heat transport (cf. section 4.2), with currents branching offshore off the EGC. Indeed, the East Icelandic Current is branching off the EGC at around 70°N (Brakstad et al., 2023) and there is a bifurcation into the shelfbreak EGC and the separated EGC at around 69°N (Våge et al., 2013; Håvik et al., 2017, see Fig. 3e).

#### 4.3.1 Greenland Sea

Two days before the MCAO, both Polar Surface Water and Atlantic-origin Water at 75°N were transported southward within the EGC (Fig. 6a,d, Fig. S3). The Polar Surface Water was covered by sea ice on the Greenland shelf, while the Atlantic-origin Water was here ventilated offshore off the sea ice edge. The ventilation of the Atlantic-origin Water along east Greenland is a recent phenomenon that has only been observed in the western Iceland Sea due to the retreat of the sea ice edge (Våge et al., 2018; Huang et al., 2021).

The mixed layer was shallower (approximately 50 m) along the EGC and deeper (300 to 600 m) at the offshore edge of the Atlantic-origin Water towards the interior of the Greenland Sea (Fig. 6a,d). Between 5°W and 0°E, in the eastern part of the basin, surface waters were warmer and more saline compared to the surface waters in the western part (see also Fig. 3a,b).

At the end of the MCAO event, and after the sea ice extent has reached its maximum (see sections 3 and 4.1), the Polar Surface Water extended farther offshore below the newly formed sea ice, preventing Atlantic-origin Water from reaching the surface (Fig. 6b,e). This explains the strong decrease in temperature (exceeding 0.5 °C) and salinity (exceeding 0.1 g·kg<sup>-1</sup>) near the surface in the area where sea ice has expanded (Fig. 6c,f). This effect is most likely occurring from 74°N northward below the sea ice in the Greenland Sea (Fig. 4a,b), as discussed in section 4.1. This result also suggests that, prior to the MCAO, ventilation of the Atlantic-origin Water occurred in the Greenland Sea along much of the EGC.

In the interior Greenland Sea, just offshore of the sea ice edge on the last day of the MCAO, the mixed layer cooled by approximately 0.05 to 0.2 °C over 6 days. This is directly related to the air-sea heat loss during the MCAO event (see section 4.2), with the mixed-layer depth increasing by up to 300 m over 6 days (Fig. 6). In the western part of the basin, the mixed-layer salinity increased (reaching  $0.01 \text{ g} \cdot \text{kg}^{-1}$ ) while it decreased ( $

**Figure 6.** Vertical cross-sections at 75°N in the Greenland Sea (cf. black line in Fig. 1). (a) Conservative Temperature (°C) before the MCAO (31/01/2020), (b) during the last day (06/02/2020), and (c) their difference. (d) Absolute Salinity (g·kg<sup>-1</sup>) before the MCAO (31/01/2020), (e) during the last day (06/02/2020), and (f) their difference. Note the non-linear scale in all panels. The thin black contours in panels (a), (b), (d), and (e) correspond to the density contours on the considered day, while in (c) and (f) they correspond to the density contours during the last day (06/02/2020). The thick black lines in panels (a), (b), (d), and (e) indicate the mixed-layer depth on the considered day, while thick blue and red lines in panels (c) and (f) indicate the mixed-layer depth before the MCAO (06/02/2020) and during the last day (06/02/2020), respectively. The magenta lines indicate areas where the ocean surface is covered by at least 15 % sea ice concentration during the considered day in panels (a), (b), (d), (e), and during the last day in (c) and (f).

the end of the MCAO should have been located below (red line, Fig. 6c,f). However, both the MLD before and at the end of the MCAO are located at greater depth than these changes. These discrepancies between the MLD and the hydrographic profiles were not present in the Iceland Sea (cf. section 4.3.2).

#### 4.3.2 Iceland Sea

245

Similarly to the Greenland Sea, fresh and cold Polar Surface Water resided below the sea ice in the boundary current region of the Iceland Sea prior to the MCAO (Fig. 7a,d). However, at 71°N, the Atlantic-origin Water along the EGC (Fig. S3) did not reach the surface and was confined below the shallow mixed layer (approximately 25 to 80 m deep). This water mass

260

265

was also warmer and more saline than in the Greenland Sea. Toward the interior of the Iceland Sea, the mixed layer became progressively warmer, more saline, and deeper (approximately 100 m to 170 m deep).

At the end of the MCAO, the Polar Surface Water extended farther offshore, also linked to the sea ice expansion (Fig. 7b,e). At the offshore edge of the sea ice and below the mixed layer, there was a shoaling of the Atlantic-origin Water. The mixed layer became even shallower in this boundary current region (approximately 25 to 80 m before and 25 m deep after the MCAO). This effect was evident up to about 16.5°W, where the Kolbeinsey Ridge is situated, marking the transition between the boundary current region to the west and the interior region to the east. In the interior Iceland Sea, the mixed layer slightly deepened by up to 30 m over the 6 days. The transition between the boundary current region and the interior coincides with the dividing line of Renfrew et al. (2023) (see their Figure 9).

Even if we identified two distinct changes in the MLD for this MCAO for the boundary current and interior regions, the mixed layer cooled and became more saline in both regions offshore of the sea ice edge (Fig. 7c,f). This is consistent with the air-sea heat loss resulting from the MCAO (see Fig. 5b). An exception to the general cooling and salinification occurred in the boundary current region at approximately 17°W, where the mixed layer cooled more than elsewhere and freshened (Fig. 7c,f), likely due to oceanic eddy activity (Fig. S3e). Apart from this area, the mixed-layer temperature decreased by approximately 0.05 to 0.2 °C over the 6 days in the Iceland Sea (Fig. 7c,f), similarly to the Greenland Sea. The salinity increased by approximately 0.002 g·kg<sup>-1</sup> to 0.2 g·kg<sup>-1</sup> within the interior Iceland Sea (Fig. 7c,f). The spatial variability in the mixed-layer response in temperature and salinity is hence large, both within the boundary current region itself, as well as within the interior region. In the boundary current region, lateral heat transport due to strong currents and eddies played a dominant role in the change of the mixed-layer properties during the MCAO (see section 4.2 and Fig. 5).

The ocean mixed-layer response to this MCAO event differs from the 28 February–13 March 2018 MCAO event, with Renfrew et al. (2023) observing a warming of the mixed layer in the boundary current region as well as a slightly deeper mixed layer after the MCAO. They attributed this warming to turbulent mixing with the warm Atlantic-origin Water that was laterally fluxed offshore from the EGC (as hypothesized by Huang et al., 2021). In contrast, we diagnose a shoaling of the Atlantic-origin Water and the mixed layer in the boundary current region, linked to a warming at the location where the mixed layer has shoaled. However, this warming remained confined below the new shallower mixed layer, with the mixed layer cooling at around 17.5 °W in the boundary current region due to air-sea heat loss. Concurrently, the Polar Surface Water was below the sea ice in our case and moved farther offshore with sea ice expansion during the event, while the Polar Surface Water was displaced onshore in Huang et al. (2021). Both, the different configurations of sea ice and water masses in the initial state of these two specific MCAO events, as well as large spatial variations in surface fluxes and lateral heat transport, most likely explain these different responses in the mixed layer.

#### 5 Conclusions

280

We used the high-resolution ocean reanalysis GLORYS12 to quantify the ocean mixed-layer response to an intense 5-day MCAO event in the Nordic Seas from 2 to 6 February 2020. In general, the ocean surface in the Nordic Seas became cooler,

Figure 7. Vertical cross-sections at 71°N in the Iceland Sea (cf. black line in Fig. 1). (a) Conservative Temperature (°C) before the MCAO (31/01/2020), (b) during the last day (06/02/2020) and (c) their difference. (d) Absolute Salinity (g·kg<sup>-1</sup>) before the MCAO (31/01/2020), (e) during the last day (06/02/2020) and (f) their difference. Note the non-linear scale in panels (c) and (f). The thin black contours in panels (a), (b), (d), and (e) correspond to the density contours on the considered day, while in (c) and (f) they correspond to the density contours during the last day (06/02/2020). The thick black lines in panels (a), (b), (d), and (e) indicate the mixed-layer depth on the considered day, while thick blue and red lines in panels (c) and (f) indicate the mixed-layer depth before the MCAO (02/02/2020) and during the last day (06/02/2020), respectively. The magenta lines indicate areas where the ocean surface is covered by at least 15 % sea ice concentration during the considered day in panels (a), (b), (d), (e), and during the last day in (c) and (f).

285

290

295

300

305

denser, and deeper throughout the MCAO event, while changes in salinity were spatially more variable. Quantifying the contribution of the air-sea heat loss in the ocean heat budget, we found that the ocean mixed layer response in the interior of the Greenland and northern Iceland Seas was dominated by air-sea exchange. In the vicinity of regions characterized by strong ocean currents and eddies, however, lateral oceanic heat transport dominated the mixed-layer response. Changes in sea ice within the marginal ice zone or along its edge also yielded enhanced variations in the mixed-layer response that were not directly attributable to the air-sea heat exchange.

We found opposite mixed-layer depth changes in the interior of the Iceland Sea and near the sea ice edge, at 71°N. Atlanticorigin Water was situated offshore off the sea ice edge in the boundary current region of the Iceland Sea, westward of the Kolbeinsey Ridge. These waters shoaled during the MCAO event, and the mixed-layer depth decreased in this area. In the interior of the Iceland Sea at 71°N, as for the Greenland Sea at 75°N, the mixed-layer depth notably increased. The newly formed mixed layers in both the interior and the boundary parts along the 71°N Iceland Sea section, as well as along the 75°N Greenland Sea section, became cooler and more saline, which can largely be attributed to the air-sea heat loss during the MCAO event.

Our analysis also revealed that ventilation of Atlantic-origin Water took place over a large part of the EGC in the Greenland Sea before the MCAO event. This phenomenon is promoted by recent sea ice retreat that potentially yields an increase in convection in the area due to enhanced air-sea heat loss (Våge et al., 2018). The 2–6 February MCAO, however, promoted an offshore expansion of the sea ice cover in the Greenland Sea, preventing ventilation during the event. GLORYS12 also suggests that the MCAO event is associated with the formation of a large sea ice structure in the western Nordic Seas, reminiscent of the Odden ice tongue (Comiso et al., 2001).

As MCAOs have a significant effect on wintertime water mass transformation in the Nordic Seas and hence the global ocean circulation, it is crucial to understand the processes that drive the spatio-temporal variability in the oceanic response to MCAOs in this region. Our results highlight the large spatial variability in the ocean mixed-layer response to a single MCAO event in the Nordic Seas. The variability stems from the complex interaction between the atmospheric forcing and the lateral ocean heat transport in conjunction with the initial ocean stratification and sea ice conditions. Although further comparison with observations is needed to assess the reliability of GLORYS12 in the Nordic Seas, this high-resolution ocean reanalysis can shed new light on the impact of MCAOs on water mass transformation in the region in a three-dimensional and time evolving spatio-temporal coverage that observations cannot provide.

Code and data availability. The ERA5 and GLORYS12 datasets used in this study are available at https://cds.climate.copernicus.eu/datasets/reanalysis-era5-single-levels?tab=overview and https://data.marine.copernicus.eu/product/GLOBAL\_MULTIYEAR\_PHY\_001\_030/services, respectively. The data were analyzed using Python (https://www.python.org/) and the GSW-Python TEOS-10 routines, developed at https://github.com/TEOS-10/GSW-Python. The codes used for processing data and making figures is available upon request.

Author contributions. SC analyzed the data and all authors contributed to writing the manuscript.

315 Competing interests. There is no competing interests.

Acknowledgements. This work was financially supported by the RCN project ARCLINK (S.C. and T.S.) and the European Union under the ERC Grant Agreement 101088452 (ROVER; S.S. and K.V.). We thank Ian Renfrew (University of East Anglia) and Anna-Marie Strehl (University of Bergen) for valuable comments on this work and insightful discussions.

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
