# Peer review of "Response of the Nordic Seas to the 2-6 February 2020 Marine Cold Air Outbreak in the GLORYS12 Ocean Reanalysis"

_EGUsphere, 2025_

## Author Comment (AC1)

**Public response to reviewers egusphere-2025-4944**

The authors thank the editor and the reviewer for their constructive comments and suggestions. Please, see below our public responses.

**Response to the reviewer**

**Reviewer 1**

**Reviewer Comment 1.1** — The authors discuss oceanic response to 2-6 February 2020 Marine Cold Air Outbreak event using atmospheric reanalysis ERA5 and ocean reanalysis GLORYS. It uses a simple water column heat budget to assess relative importance of surface heat fluxes and oceanic heat flux convergence. Major results are the temperature/salinity/MLD differences across two zonal sections in the Greenland and Iceland Sea, respectively. The authors attribute those changes to MCAO event and boundary currents. They also found "opposite MLD changes near the Ice edge compared to the interior Iceland Sea and attributed to shoaling of the Atlantic-origin water" without further explanation. The method is reasonable, yet a proper heat budget can be presented in a more rigid way. Analysis is a bit descriptive and may ignore other factors and can be strengthened with more evidence or arguments, before final publication.

**Reply**: We thank the reviewer for the constructive comments. We will address the reviewer's detailed concerns below, which reiterate the issues raised above, in the points below.

**Reviewer Comment 1.2** — Section 2.3 ocean heat budget: A proper heat budget should be evaluated in a fixed control volume, so that "Heat in" equals "Heat out" and your equation (2) holds, although a residual can occur due to modeling reason. For instance, this is performed and discussed in Årthun, M., & Eldevik, T. (2016). The referenced heat budget in Roberts et al. (2017) is integrated over mixed layer in the global ocean, probably suffer less from that. Essentially, the authors need to make sure equation (2) holds valid, if not, explain why. Could the imported cold Polar water volume from the Fram Strait be increased over this event and cools the Greenland Sea? and how to account it in the heat budget?

**Reply**: Thank you for raising this concern about the ocean heat budget. In Årthun and Eldevik (2016), the heat budget is performed over the full depth of the ocean, as done in our study, and over a specific area (control volume). They also calculate the heat flux convergence term, and obtain a residual term, which is explained by parameterized mixing in the model. In our study, we are interested in estimating the contribution of the surface heat fluxes to changes in the ocean heat content during the MCAO for every grid point (equivalent to control volumes) in the Nordic Seas. This allows us to highlight the spatial variability in the effect of the MCAO on the heat content change. Of course, it would have been advantageous to estimate the ocean heat flux convergence integrated over the vertical column for each grid point. However, given the high horizontal (less than 10 km) and relatively low temporal (24 hours) resolution of the ocean reanalysis, it is not feasible to reasonably close the ocean heat budget, because the calculation violates the CFL criterion for numerical stability. In other words, the resolved velocity, given the horizontal and temporal resolution, is less than the actual velocity of the currents.

This results in numerical instabilities and one should thus not expect to be able to close the heat budget. Hence, we rather considered the flux convergence term as a residual, because it would be the only term left, given that we integrate all the way to the ocean bottom. A potential increase in cold polar water volume transport during the event would thus be treated as a residual in our analysis. As we aim to highlight the areas where the surface heat flux term dominates the overall heat flux convergence term, i.e. where the MCAO primarily impacts the ocean heat content variability, our approach should suffice. Moreover, in addition to the challenge with the CFL criterion, reanalysis data does neither conserve mass nor energy and hence one would not expect a heat budget closure. It would thus be advisable to investigate the ability of GLORYS12 to provide an accurate estimate of the heat budget (Lellouche et al., 2021), though this is beyond the focus of our study.

**Reviewer Comment 1.3** — Section 3 Atmospheric and sea ice conditions during the MCAO: MCAOs events bring cold, dry Arctic Air over warmer open ocean, and produce strong oceanic heat loss. My impression is that eastern Nordic is under less influence of MCAOs, most studies focused on the Greenland and Iceland Seas, also as the authors do in this paper. But authors also discussed much on the eastern Nordic Sea, and attributed changes there to ocean currents there. For me, it is a bit like two regimes under two systems, the eastern warm Atlantic Water domain is subject to upstream inflow variability whereas the Greenland and Iceland Sea can be directly influenced by MCAOs. Apart from stressing this MCAO event classifies as a "very strong" intensity event, the authors probably also need to elaborate synoptic progresses or northwest-southeast contrast over the Nordic Seas. By the time the air mass from northwest reaches to the Norwegian Sea, MCAO signature is weaker and probably lost some of its convective intensity.

**Reply**: We discuss the synoptic evolution of the MCAO in section 3 (Fig. 2), where it is evident that the MCAO has different intensities in different regions (e.g. the aforementioned northwest-southeast contrast). However, this difference is less evident when looking at the overall (mean) surface heat exchange throughout the duration of the MCAO (added as Fig. 5a in the revised version of the manuscript). The overall surface heat exchange is almost as strong in the northern Norwegian Sea (eastern Nordic Seas) as in the Greenland Sea (western Nordic Seas). Regarding the two "regimes" mentioned by the reviewer, despite the surface heat exchange not being very different between the western and eastern sections of the Nordic seas, our ratio between the mean turbulent surface heat exchange and the mean ocean heat content change, discussed in section 4.2, highlights the differences in the response of the two regions.

**Reviewer Comment 1.4** — line 135-140: "Sea ice edge expanded offshore" over such a few days is not really noticeable for me at least, unless you plot ice edge lines in one figure. But you have it in Figure 4. "A structure reminiscent of the Odden ice tongue" (and thereafter) you referred is not really a "tongue", Odden Ice tongue should be a continuous tongue-like feature extending from MIZ, instead of an isolated ice pack. I believe this formed isolated sea ice is more likely due to model's deficiency, given GLORYS's insufficient resolution over this region and relatively large forcing uncertainties over MIZ in ERA5 (Renfrew et al., 2021). If you want to mention sea ice response, observational data would be more convincing. Having a quick look on ice chart from Norwegian Meteorological Institute ($https://cryo.met.no/archive/ice-service/icecharts/quicklooks/2020/20200206/denmrk\_20200206\_col.png$), I don't see this feature.

**Reply**: Thank you for this comment. We added a reference to Fig. 4 at the beginning of this paragraph. We agree that the sea ice feature does not fully resemble the Odden sea ice tongue and reworded the

description accordingly. The horizontal resolution of the reanalysis is relatively high (9 km), though the assimilation of sea ice concentration in GLORYS tends to create thicker Arctic sea ice (Lellouche et al., 2021), which might explain why the reanalysis produces this sea ice protrusion. While we agree that the model might not fully correspond to observations, we base our discussion consistently on the reanalysis and hence need to discuss our results in the light of the features in the reanalysis. An elaborate comparison of GLORYS sea ice conditions to observations at this temporal frequency is beyond the focus of our study (please see Lellouche et al. (2021) for a detailed comparison of the seasonal cycles and of the inter-annual monthly variability, between GLORYS and satellite observations).

**Reviewer Comment 1.5** — Section 4.2 Contribution of surface heat fluxes: Attributing changes in eastern AW domain to strong current and eddies but not showing convergence term due to currents explicitly (CONV in equation 2) makes interpretation and heat budget less convincing. As in Figure 5, only parts of the budgets are presented. There is a strong contrast in ocean heat tendency between the Greenland and Norwegian Sea, to emphasize atmospheric fluxes brought by MCAO and downplay ocean currents, it might be worthwhile to reconsider whether to include the Norwegian Sea under the MCAOs framework.

**Reply**: Following the reviewer's comment, we included the MCAO mean surface turbulent heat exchange (Fig. 5a) as well as its difference to ocean heat content change (Fig. 5c). Regarding the heat budget, we refer to our response further above and below, i.e., that given that we integrate to the ocean floor, the ocean heat content flux convergence can be treated as a residual to the surface forcing. The focus on the entire Nordic seas, including the Norwegian Sea, is essential for our discussion, as one of the main points is to highlight the asymmetry in the ratio (Fig. 5d) and the differing role of the surface heat exchange in modulating local ocean heat content.

**Reviewer Comment 1.6** — Figure 5(a), ocean heat content tendency is integrated from bottom to surface while MCAO event has major influence on upper ocean. Such large -3000 W/m2 heat change along the Norwegian Atlantic Slope Current probably shadows heat loss in the Greenland Sea.

**Reply**:   The reviewer has a point that one would ideally restrict this budget to the mixed layer. However, given the significant change in mixed layer depth (Fig. 4d), such a budget is challenging, as one would then need to incorporate spatial and temporal changes in mixed layer depth as well as bottom entrainment throughout the MCAO. To truthfully resolve these effects, one would ideally implement an online calculation within the model for these terms and would also need vertical velocities. However, neither of these data is available for GLORYS. Furthermore, as argued above, it is not possible to reasonably calculate horizontal ocean heat flux convergence. Hence, closed calculations for an evolving MLD are, unfortunately, not feasible. Moreover, integrating the ocean heat content down to the bottom or to near the MLD does not affect the amplitudes of the change in the western Nordic Seas, as we analyse total heat change, not specific (mass-normalised) heat exchange.

To further respond to the reviewer's concern and to justify our choice, we show that the depth chosen for the integration does not significantly affect our results when integrating down to a fixed level close to the deepest MLD (763 m, Figs. R1-1c,R1-2c) or to the bottom (Figs. R1-1d,R1-2d). However, when the vertical integral stops within the ML (109 m, Fig. R1-1a), not all atmospheric-related heat exchange is accounted for and the ratio, e.g., in the Greenland Sea, can be greater than 1 (Fig. R1-2a), meaning that the atmospheric forcing overestimates the changes in the ocean heat content. On the other hand, the ratio already features strong variability in the eastern Nordic Seas at this depth choice. When integrating to 266 m depth (Figs. R1-1b,R1-2b), the results are already qualitatively very similar to choosing 763 m (Fig. R1-1c,R1-2c) or the bottom, respectively (Fig. R1-1d,R1-2d).

Given the qualitative agreement, we argue that it is most consistent to show the analysis integrated to the bottom of the ocean, as it is mathematically the most correct representation, while retaining our interpretation of the impact of the atmospheric forcing on the ocean.

[Figure]

Figure R1-1: Mean ocean heat content tendency (W·m$^{-2}$) during the MCAO (02/02/2020-06/02/2020), at 4 different model depths: (a) 109 m, (b) 266 m, (c) 763 m, and (d) bottom depth (as in Fig. 5b of the manuscript).

**Reviewer Comment 1.7** — Figure 5(b) caption: "downward surface turbulent heat flux" is confusing, as the ocean loses heat to atmosphere, the flux is then upward.

**Reply**: Thank you for noting this, we removed "the MCAO (downward surface turbulent heat flux)" in the third and fourth lines of the caption.

**Reviewer Comment 1.8** — Section 4.3: Regarding MLD bias in GLORYS, the authors can simply calculate and correct it.

**Reply**: In principle, we agree that it would have been good to calculate the MLD. However, this is not as straightforward as the reviewer's comment implies. Due to discrete vertical resolution of GLORYS data, the MLD would also adhere to these discrete grid positions and thus not necessarily yield a more nuanced analysis compared to using the GLORYS MLD output, which is calculated online during the model integration. Furthermore, one would then also need to decide on a specific definition of the MLD

[Figure]

Figure R1-2: Ratio between mean downward surface turbulent heat flux and mean ocean heat content tendency (W·m$^{-2}$) during the MCAO (02/02/2020-06/02/2020), at 4 different model depths: (a) 763 m, (b) 902 m, (c) 1452 m, and (d) bottom depth (as in Fig. 5d of the manuscript).

(temperature, density, etc.). Hence, for consistency, and to allow for replicability of our results, we chose the MLD consistent with the reanalysis product.

**Reviewer Comment 1.9** — Regarding the upshoaling of Atlantic-origin water and mixed-layer in the boundary current region, the authors compared to the last day MCAO 06/02/2020. Does this hold if you compared to the peak day 03/02/2020? As on 06/02/2020, shown in Figure 2g, a cyclone is above the Iceland Sea centered around ice edge, with positive wind stress curl leading to ocean upwelling there, which explains upshoaling. But this cyclone is not MCAO feature as it comes from the south?

**Reply**: Thank you for the question. We clarified in the revised version of the manuscript that the changes found in the boundary region of the Iceland Sea (mixed-layer depth decrease and shoaling of Atlantic-origin water; comparing 06/02/2020 in Fig.7b,e to 31/01/2020 in Fig. 7a,d) are not due to

the MCAO, but to both cyclonic and ocean eddy activity, evident in Fig. 5d (revised version). From 02/02/2020 to 05/02/2020, the cyclonic activity was less intense, but the mixed-layer depth kept decreasing concomitantly with an intensification of the shoaling of the Atlantic-origin water (Fig. R1-3 and Fig. R1-4 below).

[Figure]

Figure R1-3: As Fig. 7a,b in the manuscript, but for each day between 30 January 2020 and 9 February 2020.

[Figure]

Figure R1-4: As Fig. 7c in the manuscript, but for each day between 1 January 2020 and 9 February 2020.

**Reviewer Comment 1.10** — Line 16: I am not sure (Gebbie and Huybers, 2010) is a proper reference here.

**Reply**: Thank you. We changed the reference to "Chafik and Rossby (2019)", which is a more appropriate reference indicating that the Nordic Seas play an important role in the AMOC.

**Reviewer Comment 1.11** — Line 20: Overflow occurs not only via Denmark Strait but also via eastern past of Greenland-Scotland ridge.

**Reply**: We rewrote the sentence and clarified that Denmark Strait is only one of the overflows across the Greenland-Scotland Ridge.

**Reviewer Comment 1.12** — Lines 50-55: the introduction of GLORYS here could be integrated in section 2.1 Data and Methods.

**Reply**: While we can relate to the suggestion, we believe that introducing GLORYS in the introduction helps to underline our choice for this dataset, given that it provides an opportunity to study the mesoscale impacts of MCAOs on the ocean in the Nordic Seas.

**Reviewer Comment 1.13** — Figure 1: Some arrowheads are disconnected.

**Reply**: Thank you for pointing this out, the figure has been modified.

**Reviewer Comment 1.14** — Line 66-67: Given forcing uncertainties, authors should be more cautious on later interpretation near MIZ.

**Reply**: Thank you. See the response to comment 1.4.

**Reviewer Comment 1.15** — Line 124: Fram Strait.

**Reply**: Corrected.

**Reviewer Comment 1.16** — Line 128: What "flow configuration" ?

**Reply**: Modified to "The wind conditions" that were described in the beginning of the paragraph ("north-westerlies").

**Reviewer Comment 1.17** — Line 154-155: It seems redundant, basically, you are saying: GLORYS is consistent with (the combination of GLORYS + satellite observation), and we already know that GLORYS has assimilated satellite observation.

**Reply**: We wanted to refer to the figure 1 of Huang et al. (2023), but we agree that this comparison is not adequate, and therefore have removed it from the manuscript.

**Reviewer Comment 1.18** — Lines 190-194: supporting arguments or references? related to major comments.

**Reply**: The discussion in these lines is based on our figures. Hence, it is not clear what references or supporting arguments the reviewer deems needed in this context. We added a reference to Fig. 3e as well, when mentioning the strong currents and associated eddies in this paragraph.

**Reviewer Comment 1.19** — Figure 6: incorrect captions: "before the MCAO (06/02/2020) and during the last day (06/02/2020)" and also Figure 7 caption: "before the MCAO (02/02/2020) and during the last day (06/02/2020)"

**Reply**: Corrected, thank you.

**Reviewer Comment 1.20** — Line 294: can becoming more saline be attributed to heat loss?

**Reply**: Changes in salinity can be attributed to different factors, rendering a direct attribution difficult. Hence, we removed "and more saline" in this sentence.

**References**

Årthun, M. and Eldevik, T.: On anomalous ocean heat transport toward the Arctic and associated climate predictability, Journal of Climate, 29, 689–704, 2016.

Chafik, L. and Rossby, T.: Volume, heat, and freshwater divergences in the subpolar North Atlantic suggest the Nordic Seas as key to the state of the meridional overturning circulation, Geophysical Research Letters, 46, 4799–4808, 2019.

Huang, J., Pickart, R. S., Chen, Z., and Huang, R. X.: Role of air-sea heat flux on the transformation of Atlantic Water encircling the Nordic Seas, Nature Communications, 14, 141, https://doi.org/10.1038/s41467-023-35889-3, 2023.

Lellouche, J.-M., Greiner, E., Bourdallé-Badie, R., Garric, G., Melet, A., Drévillon, M., Bricaud, C., Hamon, M., Le Galloudec, O., Regnier, C., et al.: The Copernicus Global 1/12° Oceanic and Sea Ice GLORYS12 Reanalysis, Frontiers In Earth Science, 9, https://doi.org/10.3389/feart.2021.698876, 2021.

---

## Author Comment (AC2)

**Public response to reviewers egusphere-2025-4944**

The authors thank the editor and the reviewer for their constructive comments and suggestions. Please, see below our public responses.

**Response to the reviewer**

**Reviewer 2**

**Reviewer Comment 2.1** — This paper examines how the Nordic Seas responded to a significant marine cold air outbreak in the winter of 2020. The authors use atmospheric and oceanic reanalysis data to examine this question. The authors quantify the contributions of the mean surface turbulent heat flux relative to the mean change in ocean heat content during the event. The authors find the air-sea fluxes dominant in the western part of the Nordic Seas, while finding lateral oceanic heat transport more important in the east. This is an interesting paper, looking in detail at a given atmospheric event. All possible processes are considered, and their relative roles (spatially) are considered and explored. The paper is generally well written. That said, there are some limitations to the study, and some places where the work could be expanded on. Thus, I would recommend moderate revisions. Specific comments are provided below.

**Reply**: We thank the reviewer for the comments and suggestions for improvement provided for the manuscript.

**Reviewer Comment 2.2** — The ocean fields used come from the GLORYSV12 reanalysis product. The authors reference other studies showing the quality of this product in the Nordic Seas. That said, it is a single product, with biases. Why did the authors not consider using several reanalyzes to determine how robust their analysis is?

**Reply**: The reviewer has a fair point and it would have been interesting to compare the response in GLORYS to other ocean reanalyses that provide output at a daily or higher temporal resolution. However, reanalysis such as ORAS5 and RARE only provide monthly and 5-day temporal resolution, respectively. GLORYS12 is thus superior in terms of resolution compared to most reanalysis products. Given that our study focuses on the oceanic response to a specific MCAO, the restriction to one reanalysis product is justified and a detailed reanalysis inter-comparison would be beyond the focus of our study and change the character of this manuscript from a physical discussion of the oceanic response to a discussion of a model inter-comparison.

**Reviewer Comment 2.3** — As a side point, at 1/12 degree resolution, the GLORSYSV12 product is not fully eddy-permitting. It resolves the largest eddies in the region, but not all. I believe the terminology eddy-rich would be more appropriate.

**Reply**: Thank you for pointing this out. We agree that not all the eddies are fully resolved in the Nordic Seas in GLORYS12 (e.g. size less than $\sim$ 5 km). However, the terminology "eddy-resolving" is used in

the GLORYS12 overview paper (Lellouche et al., 2021) and we consistently adopted this terminology in our manuscript.

**Reviewer Comment 2.4** — The atmospheric product used is ERA5. It has known biases, including being too warm in polar regions, and having events with extreme wind anomalies. It would be good to confirm how those might impact the authors' study for the given MCAO.

**Reply**: Renfrew et al. (2021) compared ERA5 to observations for winter 2018 over the Iceland and Greenland Seas and found that ERA5 performs well over the ice-free ocean, with biases "significantly less than the observed standard deviations for all variables" (T2m, wind speed, wind direction, and turbulent heat fluxes). Moreover, the observed wind speed during the MCAO in the COMBLE campaign (see supplement in Geerts et al., 2022) is consistent with ERA5 on 2 and 4 February (max 15 m/s) at two different sites in the eastern Nordic Seas (Bear Island and Andenes). However, ERA5 has been shown to have some limitations over the marginal ice zone. We added this discussion in section 2.1 and section 3 (see also comment 2.7).

**Reviewer Comment 2.5** — The methods section requires some further details on the approaches used. The authors discuss using fields from ERA5. But what bulk formulae were used in the calculations? How were these computed in conjunction with GLORYSV12 – especially if the authors used daily outputs from the ocean reanalysis and higher frequency fields from ERA5? How were the fluxes computed over the model grid cells with sea-ice? Were all the calculations done at the location of the model T-grid cells? Or interpolated to them?

**Reply**: We did not use any bulk formulae, as we used the fluxes provided by ERA5 for our analysis (3-hourly surface sensible and latent heat fluxes, cf. section 2.1). We used ERA5, as GLORYS12 is forced by ERA5 after 2019 (cf. section 2.1). In GLORYS12, momentum and heat turbulent surface fluxes are computed from the Large and Yeager (2009) bulk formulae, where atmospheric quantities were sampled at a 3-hour resolution and the calculations were done on the model Arakawa C-grid. See Lellouche et al. (2021) for details on production of the GLORYS12 reanalysis.

**Reviewer Comment 2.6** — I don't understand why the heat budget is computed over the total ocean depth (i.e. using Dbot). The lower layers are not going to be impacted by the atmospheric forcing. Thus, the size of the signals shown will be impacted by the ocean depth, which doesn't seem relevant to the authors' questions. Especially for the ratio calculation, such as in figure 5. Yes, not using the bottom depth does mean looking at vertical heat fluxes, but it shouldn't be especially hard to compute. As well, that approach could mean that the authors could look at the very fluxes into and out of given watermass layers, such as the Atlantic Water layer.

**Reply**: Thank you for raising this concern. As mentioned in the response to reviewer 1 (cf. reply to comment 1.6, Fig. R1-1 and Fig. R1-2 which is the same as Fig. R2-1 below), integrating the ocean heat content down to the bottom or to near the MLD does not really affect the amplitudes of the change in the western Nordic Seas, as we analyse total heat change, not specific (mass-normalised) heat exchange. Integrating to the MLD would imply that we would integrate to a spatially and temporally varying depth across the domain of interest, requiring further terms in the heat budget equation, such as entrainment and heat content changes due to the change of the MLD. To truthfully resolve these effects, one would ideally implement an online calculation within the model. As indicated by the reviewer, when

integrating down to a fixed level close to the deepest MLD (e.g., 763 m, Fig. R1-2c), the residual would also include contributions from the vertical heat flux convergence, which we, however, cannot calculate as GLORYS does not provide vertical velocities. Furthermore, as mentioned in the reply to comment 1.2, it is not feasible to provide reasonable estimates of the lateral ocean heat flux convergence. To justify our choice, we show that the depth chosen for the integration does not significantly affect our results when integrating down to a fixed level close to the deepest MLD or to the bottom (compare Fig. R2-1c and Fig. R2-1d below). However, when the vertical integral stops within the ML (109 m), the ratio, e.g., in the Greenland Sea, can be greater than 1, meaning that the atmospheric forcing overestimates the changes in the ocean heat content (Fig. R2-1a), while the ratio still features strong variability in the easter Nordic Seas. Already when integrating to 266 m depth (Fig. R2-1b), the results are qualitatively very similar to chosing 763 m or the bottom, respectively (Fig. R2-1c,d).

[Figure]

Figure R2-1: Ratio between mean downward surface turbulent heat flux and mean ocean heat content tendency (W·m$^{-2}$) during the MCAO (02/02/2020-06/02/2020), at 4 different model depths: (a) 763 m, (b) 902 m, (c) 1452 m, and (d) bottom depth (as in Fig. 5d of the manuscript).

**Reviewer Comment 2.7** — Given the importance of sea-ice in modifying the air-sea fluxes, I would have liked to seen more discussed about it in the paper (rather than just seeing the ice edge on figures). Since the authors discuss this at line 140 (for example), could the authors' show sea-ice growth rates, thickness changes and/or advection during the MCAO event, as the sea-ice is likely also responded to it.

**Reply**: Thank you for raising this interesting question about the sea ice response. Unfortunately, GLORYS12 does neither provide sea ice growth rates nor sea ice advection. However, we refer to the sea ice volume changes during the MCAO in line 138 and refer to Fig. S1 and added "likely indicating a combination of south-eastward sea ice advection and sea ice formation at the edge."

**Reviewer Comment 2.8** — In section 4.3.1, the authors' discuss how both Polar Surface Water and Atlantic origin Water were transported southward. It would be nice to quantify this, with timeseries, to help see how the transport changes during the MCAO.

**Reply**: Thank you for the suggestion. While we do not quantify the actual volume or heat transport (nor heat flux convergence in general, see our reply to comment 1.2 to reviewer 1), the strength of the current is visible in Fig. S3b,e (figure cited in the beginning of sections 4.3.1 and 4.3.2). When comparing the sea surface velocity before and at the end of the MCAO, we see that there is only a small change in the 75°N section in the Greenland Sea (cf. Fig. R2-2 below), and the Polar Surface Water and Atlantic-Origin Water velocities within the EGC during the MCAO are therefore relatively constant. In the Iceland Sea at the 71°N section (cf. Fig. R2-3), a distinct change in velocity is visible, though this change is related to an eddy, as mentioned in the manuscript (section 4.3.2. Please see also comment 1.9). In our framework, any change in transport is taken into account as a change in ocean heat flux convergence (accounted for as a residual), and we do not assess how it might have been altered due to the MCAO, as this would require a detailed momentum analysis, which is beyond the scope of this manuscript.

[Figure]

Figure R2-2: Vertical cross-sections of mean sea water speed (cm·s⁻¹) at 75°N in the Greenland Sea, before the MCAO (31/01/2020, first row) and at the end of the MCAO (06/02/2020). The northward components are shown in (b) and (e), and the eastward components are shown in (c) and (f). The thick black lines indicate the mixed-layer depth. The magenta lines indicate the sea ice extent. The thin black contours correspond to the density contours.

[Figure]

Figure R2-3: Vertical cross-sections of mean sea water speed (cm·s⁻¹) at 71°N in the Iceland Sea, before the MCAO (31/01/2020, first row) and at the end of the MCAO (06/02/2020). The northward components are shown in (b) and (e), and the eastward components are shown in (c) and (f). The thick black lines indicate the mixed-layer depth. The magenta lines indicate the sea ice extent. The thin black contours correspond to the density contours.

**Reviewer Comment 2.9** — In the conclusion, the authors start by stating they use the high-resolution GLORYSV12 reanalysis. Beyond my above point related to eddy-permitting vs eddy-rich, I don't feel like the authors really take advantage of the resolution of the product. It would be good to know where the given resolution is most helpful for the study. Or, even better, can the authors examine the role of the eddies and mesoscale in their results, for example, their role in the lateral oceanic heat transport in the eastern part of the domain.

**Reply**: This is a very interesting and relevant question. However, to quantify the role of the mesoscale requires comparing our GLORYS analysis to another dataset at coarser resolution that does not resolve the scales inherent to GLORYS, with the disadvantage that difference will be due to both resolution as well as model version. Ideally, one would need a dataset driven by the same forcing using the same model at different resolution to prove the relevance of scales. Alternatively, one could perform a filtering, removing certain scales from the data, but such an analysis would also suffer from certain shortcomings, as all results will mainly be a result of the choice of filtering. Most likely, the main difference when using different ocean resolutions would be the representation of the currents and eddies. From our analysis, it is evident that the strength and presence of currents and eddies significantly alters the visibility of the oceanic response to the atmospheric forcing. In coarser resolution ocean models, eddies would be absent and currents would be significantly weaker in strength, which would then most likely yield weaker variability across the Nordic Seas in the response to the atmospheric forcing. This discussion is already implicit in our manuscript, but we tried to further clarify these aspects in the conclusions.

**Reviewer Comment 2.10** — L124: Fram Strait

**Reply**: Corrected, thank you.

**Reviewer Comment 2.11** — Figure 4 caption: The authors state they are comparing surface ocean conditions pre and post the MCAO. Yet, if the MCAO covered Feb 2-6, while Jan 31 is pre-MCAO, Feb 6 is still during the MCAO. Shouldn't the plots then use Feb 7? And why not Feb 1, instead of two days before for the pre-MCAO panels?

**Reply**: Thank you for noting this. We corrected "the post-MCAO (06/02/2020)" to "end-MCAO" as it is written for the sea ice edges in the end of the caption. We also chose to show February 6th in the manuscript and not Feburary 7th, as the effects on the ocean are the same (for example regarding the Conservative Temperature differences, compare Fig. R2-4a and b below). In this way, the figures are consistent with Fig. 2 (last row), showing the atmospheric conditions throughout the MCAO. Similarly, using January 31st or Feburary 1st does not really affect the result here (Fig. R4-2).

[Figure]

Figure R2-4: Conservative Temperature difference at different starting and end dates. (a) 06/02/2020 - 31/01/2020, as in Fig. 4a in the manuscript. (b) 07/02/2020 - 31/01/2020. (c) 06/02/2020 - 01/02/2020. (d) 07/02/2020 - 01/02/2020

**Reviewer Comment 2.12** — Figures 6 and 7: The thick black, blue and red lines to show the mixed layer depth are discontinuous, and not easy to follow in places where the depth changes rapidly. The plotting of these lines could be improved (with a continuous line).

**Reply**: Thank you for the suggestion. We changed the lines representing the mixed-layer depths into continuous lines.

**References**

Geerts, B., Giangrande, S. E., McFarquhar, G. M., Xue, L., Abel, S. J., Comstock, J. M., Crewell, S., DeMott, P. J., Ebell, K., Field, P., et al.: The COMBLE Campaign: A study of marine boundary layer clouds in Arctic cold-air outbreaks, Bulletin of the American Meteorological Society, 103, E1371–E1389, https://doi.org/10.1175/BAMS-D-21-0044.1, 2022.

Large, W. and Yeager, S.: The global climatology of an interannually varying air–sea flux data set, Climate dynamics, 33, 341–364, 2009.

Lellouche, J.-M., Greiner, E., Bourdallé-Badie, R., Garric, G., Melet, A., Drévillon, M., Bricaud,

C., Hamon, M., Le Galloudec, O., Regnier, C., et al.: The Copernicus Global 1/12° Oceanic and Sea Ice GLORYS12 Reanalysis, Frontiers In Earth Science, 9, https://doi.org/10.3389/feart.2021.698876, 2021.

Renfrew, I. A., Barrell, C., Elvidge, A., Brooke, J., Duscha, C., King, J., Kristiansen, J., Cope, T. L., Moore, G. W. K., Pickart, R. S., et al.: An evaluation of surface meteorology and fluxes over the Iceland and Greenland Seas in ERA5 reanalysis: The impact of sea ice distribution, Quarterly Journal of the Royal Meteorological Society, 147, 691–712, https://doi.org/0.1002/qj.3941, 2021.